# Time-Course of Transcriptomic Change in the Lungs of F344 Rats Repeatedly Exposed to a Multiwalled Carbon Nanotube in a 2-Year Test

**DOI:** 10.3390/nano13142105

**Published:** 2023-07-19

**Authors:** Motoki Hojo, Ai Maeno, Yoshimitsu Sakamoto, Yukio Yamamoto, Yuhji Taquahashi, Akihiko Hirose, Jin Suzuki, Akiko Inomata, Dai Nakae

**Affiliations:** 1Department of Pharmaceutical and Environmental Sciences, Tokyo Metropolitan Institute of Public Health, 3-24-1 Hyakunincho, Shinjuku-ku, Tokyo 169-0073, Japan; 2Division of Cellular and Molecular Toxicology, Center for Biological Safety and Research, National Institute of Health Sciences, 3-25-26 Tono-machi, Kawasaki-ku, Kawasaki 210-9501, Kanagawa, Japan; 3Chemicals Assessment and Research Center, Chemicals Evaluation and Research Institute, Japan, 1-4-25 Koraku, Bunkyo-ku, Tokyo 112-0004, Japan; 4Department of Medical Sports, Faculty of Health Care and Medical Sports, Teikyo Heisei University, 4-1 Uruido-Minami, Ichihara 290-0193, Chiba, Japan

**Keywords:** carbon nanotubes, lung cancer, transcriptomics, inflammation, intratracheal instillation, MWNT-7, F344 rats

## Abstract

Despite intensive toxicological studies of carbon nanotubes (CNTs) over the last two decades, only a few studies have demonstrated their pulmonary carcinogenicities in chronic animal experiments, and the underlying molecular mechanisms are still unclear. To obtain molecular insights into CNT-induced lung carcinogenicity, we performed a transcriptomic analysis using a set of lung tissues collected from rats in a 2-year study, in which lung tumors were induced by repeated intratracheal instillations of a multiwalled carbon nanotube, MWNT-7. The RNA-seq-based transcriptome identified a large number of significantly differentially expressed genes at Year 0.5, Year 1, and Year 2. Ingenuity Pathway Analysis revealed that macrophage-elicited signaling pathways such as phagocytosis, acute phase response, and Toll-like receptor signaling were activated throughout the experimental period. At Year 2, cancer-related pathways including ERBB signaling and some axonal guidance signaling pathways such as EphB4 signaling were perturbed. qRT-PCR and immunohistochemistry indicated that several key molecules such as Osteopontin/Spp1, Hmox1, Mmp12, and ERBB2 were markedly altered and/or localized in the preneoplastic lesions, suggesting their participation in the induction of lung cancer. Our findings support a scenario of inflammation-induced carcinogenesis and contribute to a better understanding of the molecular mechanism of MWCNT carcinogenicity.

## 1. Introduction

Carbon nanotubes (CNTs) have been used in assorted disciplines such as electronics, energy storage, materials science, and medicine, which raises a public health concern about the toxicity of CNT exposure. A mounting number of in vivo and in vitro studies over the past two decades have documented the toxicities of CNTs, including the induction of acute inflammation, chronic inflammation accompanied by fibrosis, and cancer in pulmonary and mesothelial tissues (reviewed in [1,2,3,4,5]). In particular, the chronic toxicities of CNTs with a needle-like structure were intensively evaluated because their shape and physical and chemical durability may be similar to asbestos [6,7].

Intraperitoneal injection studies revealed that a thick, long, needle-like multiwalled carbon nanotube (MWCNT), MWNT-7 (also known as Mitsui-7, MWCNT-7, and XNRI-7), induced mesotheliomas in *p53* heterozygous mice and Fischer 344 rats [8,9]. Based primarily on the intrascrotal injection study by Sakamoto et al. [9], one intraperitoneal injection study with rats by Nagai et al. [10], a second intraperitoneal injection study using *p53* heterozygous mice by Takagi et al. [11], and an initiation-promotion study in mice in which lung cells were initiated with 3-methylcholanthrene followed by exposure to MWNT-7 by whole-body inhalation for 15 days by Sargent et al. [12], MWNT-7 was classified as a Group 2B carcinogen by the International Agency for Research on Cancer (IARC) in 2014 [13]; the final report was published in IARC monograph 111 in 2017 (Monograph 111 Table 3.2 (page 70) states that NT50a is MWCNT-7). MWNT-7 was subsequently found to induce cancer in the rat lung in a 2-year inhalation study [14]. This inhalation study established that MWNT-7 is a complete carcinogen in rats. Thus, the most important endpoint of CNTs’ toxicity is carcinogenicity, and carcinogenicity should be employed for the risk assessment of CNTs. There is, however, a paucity of experimental evidence regarding the carcinogenicity or genotoxicity of other types of CNTs. Given the tremendous variety of CNT species that have been and are being developed (e.g., CNTs with differences in the number of walls, lengths, and surface modifications), the implementation of 2-year carcinogenicity tests for each CNT may be unrealistic.

Particle and fiber toxicologists have begun focusing on toxicogenomic analysis as a complementary approach to traditional chronic toxicity tests. These approaches can provide a mechanistic understanding of toxicities as well as biomarker candidates that may be applicable in animal experiments, in vitro tests, and occupational exposure assessments. For example, time-course analyses of mRNA expression in the lung tissues of rodents exposed to MWCNT or single-walled carbon nanotube (SWCNT) identified gene sets associated with inflammation and fibrosis in the lung [15,16,17]. More recently, to explore useful blood biomarkers, Khaliullin et al. performed a comparative analysis of the transcriptome of mRNA in the lung and blood of mice exposed to MWCNTs by pharyngeal aspiration [18]. Comparative analyses of lung and blood mRNA and miRNA from mice exposed to MWCNT by pharyngeal aspiration with human clinical data and in vitro MWCNT-exposed human cells identified possible markers for occupational surveillance and human cancer risk and prognosis [19,20]. Shvedova et al. evaluated mRNA and ncRNA in the blood of workers in a CNT factory to search for potential biomarkers for monitoring MWCNT exposure in humans [21]. In addition, researchers applied this powerful tool for grouping and classifying various test materials [22,23,24]. Most of these omics readouts demonstrated that CNT exposure resulted in a similar pattern of enrichment of genes involved in pathways and networks associated with inflammation, reactive oxygen species (ROS) generation, fibrosis, DNA damage, and cell proliferation. However, omics studies with relatively short-term animal experiments or human exposures primarily addressed inflammatory and fibrotic endpoints. Thus, comprehensive molecular analyses of CNT-induced carcinogenesis are needed for a better understanding of whether inflammation is a main biological response even in long-term experiments, whether prolonged inflammation does result in tumorigenesis, and what types of pathways are perturbed before and during cancer development.

The goal of this study is to explore the transcriptomic landscape of lung tumor development induced by CNTs in rats, which can provide a hint to evaluate CNTs with the carcinogenic endpoint in toxicogenomic analyses. Until now, only one study has proven the lung carcinogenicity of a CNT (MWNT-7) using a 2-year inhalation test [14]. However, a series of 2-year studies using intratracheal spraying demonstrated the carcinogenic potential of some other types of CNTs, including MWCNT-N, MWCNT-B, and double-walled carbon nanotubes (Tocana) [25,26,27,28]. Also, we have recently demonstrated that 2-year intermittent exposures to MWNT-7 by intratracheal instillation induced lung adenomas and adenocarcinomas, as well as pleural mesotheliomas, in rats [29]. In this study, using lung samples dissected from rats necropsied at 0.5, 1, and 2 years after the beginning of the experiment, we performed an RNA-seq-based transcriptomic analysis. Significantly differentially expressed genes were identified at each time point. Ingenuity Pathway Analysis revealed an overrepresentation of inflammation-related biological functions and pathways at all three time points, and in the 2-year samples, there was a marked perturbation of gene expression involved in cancer development.

## 2. Materials and Methods

### 2.1. MWCNT

MWNT-7 (also known as Mitsui-7, MWCNT-7, and XNRI-7; lot, 060 125-01 k) was kindly donated by Mitsui & Co., Ltd. (Tokyo, Japan). The bulk materials were pretreated by filtration using a 53-μm mesh, which removed the agglomerates and aggregates of fibers without altering the size distribution of the fibers, and a critical point drying technique [30]. The MWCNT was then baked at 200 °C for 2 h in a dry heat sterilizer for the elimination of endotoxin. The MWCNT was suspended in sterile 0.9% saline containing 0.1% Tween 80. The morphology of most MWNT-7 fibers in the suspension was straight, and the average length and width of the fibers were 5.11 μm and 84.7 nm, respectively (Appendix A). The sample was fully characterized in our previous reports [29,30,31].

### 2.2. Animals and Experimental Design

In the present study, all lung samples used for transcriptome analysis were sourced from a previous 2-year study [29]. The conditions of the animal experiment are briefly described as follows. Five-week-old malespecific pathogen-free Fischer 344 (F344/DuCrlCrlj) rats were purchased from Jackson Laboratories Japan (Kanagawa, Japan). The rats were housed in a polycarbonate cage (3 rats per cage) with autoclaved paper bedding (Alpha-Dri, Shepherd Specialty Papers, Watertown, TN, USA). The rats were maintained in a room at a temperature of 23 ± 0.1 °C and 53.1 ± 7.9% relative humidity on a 12 h light–dark photophase cycle, and given a standard basal diet (CE-2, CLEA Japan, Tokyo, Japan) and drinking water via a bacterial filter ad libitum.

The study consisted of three groups: the vehicle control group, the low-dose group, and the high-dose group, which administered MWNT-7 at 0, 0.125, and 0.5 mg/kg body weight, respectively, once every four weeks (Figure 1). The rats were deeply anesthetized by inhalation of 3% isoflurane (Pfizer, New York, NY, USA), held on a holder, and then the vehicle or the MWCNT suspension was instilled through the larynx into the lung using a feeding cannula connected to a syringe. From 9 weeks of age, intratracheal administration of MWCNT to the rats was performed 26 times at intervals of 4 weeks for 2 years (Figure 1), which resulted in mean lung burdens of 0.9 and 3.6 mg/lung for the low-dose group and high-dose group, respectively, at the terminal necropsy [29]. The general condition of the rats was monitored twice daily, and body weights were measured every week. At the interim sacrifices and termination of the study, animals were killed by exsanguination through the abdominal aorta under 3% isoflurane anesthesia.

Thirty animals in each group were histopathologically examined for carcinogenicity at 104 weeks after the beginning of the experiment. Satellite animals were also sacrificed at weeks 26 (Year 0.5), 52 (Year 1), and 104 (Year 2) to examine lung burden and pleural lavage fluid. Results of the histopathological analysis demonstrated that the incidence of lung tumors was dose-dependently increased and was significantly higher in the high-dose group compared with the control group (Figure 1). In regard to proliferative lesions, only reactive hyperplasias of alveolar cells were observed until Year 1, while atypical hyperplasias, adenomas, and adenocarcinomas were frequently found at Year 2 (Appendix A). Details of methods and results of carcinogenicity of lung and mesothelioma were previously described [29]. In the present study, we analyzed RNA from the lungs of 15 rats in the control and high-dose groups (Figure 1).

This study was conducted at the Tokyo Metropolitan Institute of Public Health according to the Animal Research Reporting of In Vivo Experiments (ARRIVE) guideline [32]. The facilities are accredited by the Japan Pharmaceutical Information Center. The experimental protocol was approved by the Animal Experiment Committee of the Tokyo Metropolitan Institute of Public Health (approval numbers of animal experiments in this study: 30–26, 19–24, and 20–22).

### 2.3. RNA Preparation and Sequencing

To limit the effects of circadian influence, rats used for molecular analyses in each group were necropsied at 9–12 a.m. All tissues containing tumor nodules recognized by gross examination at necropsy were excluded from RNA analysis. Small portions of tissue were dissected from lung accessory lobes, snap-frozen in the presence of liquid nitrogen, and stored at −80 °C. The remnant accessory lobe was inflated with 10% neutral buffered formalin under positive pressure using a syringe and processed for histopathology. Fixed tissues were embedded in paraffin, sectioned (4 μm thickness), stained with hematoxylin and eosin (H&E), and examined microscopically for confirmation of the absence of tumors. Tumor-free frozen tissue samples were thoroughly ground at 2200 rpm for 15 s using a grinder (Multi-beads Shocker^®^; Yasui Kikai, Osaka, Japan) and then processed for total RNA extraction by RNeasy Mini Kit as per the manufacturer’s protocol (74104, QIAGEN, Venlo, The Netherlands). Each total RNA sample was quantified by a spectrophotometer (Nanodrop One; Thermo Fisher Scientific, Waltham, MA, USA), and the quality of the RNA was assessed with a 2200 TapeStation (Agilent Technologies, Santa Clara, CA, USA). Samples from the 15 control rats and the 15 rats exposed to MWNT-7 passed the quality criterion of an RNA integrity number (RIN) > 7.0. Nine samples from the controls and nine samples from the MWNT-7-exposed rats (N = 3 from each time point) were randomly chosen for subsequent processing (Figure 1). All 15 samples from the controls and the MWNT-7-exposed rats were used for qRT-PCR. RNA-seq libraries were prepared using an NEBNext Ultra II RNA Library Prep Kit for Illumina (New England Biolabs, Ipswich, MA, USA). Equimolar amounts of barcoded libraries were pooled and sequenced on a NovaSeq 6000 Sequencing System (Illumina, San Diego, CA, USA).

### 2.4. Data Analysis

RNA-Seq reads were imported into CLC Genomics Workbench software (version 22.0.2, QIAGEN) and processed for further analyses. All reads were batch processed and mapped to the *Rattus norvegicus* reference genome. Raw expression data were normalized using the trimmed mean of the M-values normalization method (TMM normalization) and presented as TMM-adjusted Counts Per Million (CPM). Significantly differentially expressed genes (DEGs) were identified as meeting the following criterion: false discovery rate (FDR) < 0.05 and |Log_2_ Fold Change| > 1. Principal components analysis (PCA) was performed on the full set of genes from the CPM data. All DEGs or DEGs involved in target canonical pathways were hierarchically clustered using Euclidean distance and complete linkage. The RNA-seq datasets were deposited in the NCBI Gene Expression Omnibus database (https://www.ncbi.nlm.nih.gov/geo/query/acc.cgi?acc=GSE234151, accessed on 8 June 2023).

Ingenuity Pathway Analysis (IPA) (QIAGEN) was used to identify relevant biological processes, functions, and pathways from the expression data. Core Analysis was performed on a dataset based on the Log_2_ Fold Change for DEGs, which generated lists of significant Diseases and Biofunctions, Canonical Pathways, Upstream Regulators, and Regulator Effects. Regulator Effects analysis connects an upstream regulator to a regulated gene set and a particular phenotypic or functional outcome. A Graphical Summary is an overview of the major biological themes in the Core Analysis and illustrates how these concepts relate to one another using each predicted entity such as Canonical Pathways, Upstream Regulators, and Diseases and Biofunctions. For each analysis, a significance value was calculated based on a right-tailed Fisher’s exact test by the IPA internal system: −log_10_ (*p*-value) > 1.3 was considered statistically significant (*p* < 0.05). The z-score is a statistical measure of how closely the actual expression pattern of molecules in the dataset compares to the pattern that is expected based on the QIAGEN Biomedical Knowledge Base. In this study, pathways given a z-score > 1 were depicted as positive (activated) pathways, while those given a z-score < −1 were depicted as negative (inhibited) pathways. A detailed description of IPA is shown on the QIAGEN website (https://qiagen.my.salesforce-sites.com/KnowledgeBase/KnowledgeNavigatorPage?categoryName=IPA, accessed on 8 June 2023). Venn diagrams were generated using the Bioinformatics and Evolutionary Genomics website (http://bioinformatics.psb.ugent.be/webtools/Venn/, accessed on 15 April 2023).

### 2.5. Quantitative Real-Time Polymerase Chain Reaction (qRT-PCR)

Total RNA samples were obtained from all 15 rats as described in Section 2.3. The first-strand cDNA library was synthesized from 1 μg of total RNA by using SuperScript III First-Strand Synthesis Super Mix (Thermo Fisher Scientific, Waltham, MA, USA). qRT-PCR assays were performed by using an SYBR Green Master Mix (Thermo Fisher Scientific) and gene-specific primer sets (Appendix A). A 7500 Fast Real-time PCR System (Thermo Fisher Scientific) was used for the assay with the following program: holding stage, 95 °C per 10 min; cycling stage (40 cycles), 95 °C × 10 s–60 °C × 1 min; and melting stage, 95 °C × 15 s–60 °C × 1 min–95 °C × 30 s–60 °C × 15 s. Changes in gene expression relative to the different samples were calculated using the value of a housekeeping gene, *Gapdh,* according to the standard 2^−ΔΔCt^ method. Each result is shown as fold change relative to the control group at Year 0.5, except for 2 genes, *Unc5d* and *Cav1*, which were not or nearly not detected at Year 0.5. *Unc5d* and *Cav1* data are shown relative to the control group at Year 1.

### 2.6. Histopathology

For immunohistochemical analysis of DEGs, histological slides were obtained from the main 2-year study [29], in which lungs were fixed in 10% neutrally buffered formalin at 30 cm H_2_O pressure and routinely processed for paraffin-embedded histological specimens. Antigen retrieval was performed in Tris-EDTA buffer (pH 9.0; 415211, Nichirei, Tokyo, Japan), followed by the inactivation of endogenous peroxidase by immersion in H_2_O_2_. After blocking with Protein Block (X0909; Agilent Technologies, Santa Clara, CA, USA) for 20 min at room temperature, the sections were treated with primary antibodies (Appendix A) for 1 h at room temperature. Horseradish peroxidase-secondary antibody conjugate (K4061, Agilent Technologies) was used for the detection of diaminobenzidine signals, as per the manufacturer’s instructions.

### 2.7. Statistics

Statistical analysis of qRT-PCR values was performed using the Student’s *t*-test (GraphPad Prism 9 (version 9.5.1): GraphPad Software, Boston, MA, USA) for comparisons between MWCNT-treated groups and time-matched control groups. Statistical significance was set at *p* < 0.05.

## 3. Results

### 3.1. Identification of Significantly Differentially Expressed Genes (DEGs)

PCA was performed on the full set of RNA-Seq reads. The first two principal components (PCs) captured close to 50% of the variance of the data set. The PC 2 axis formed a partition between the two experimental groups (Figure 2A). Individual rats in the MWCNT-treated groups were positioned along the PC 1 and PC 3 axes, forming clusters according to time points. No clear separation according to time points was observed for the control animals (Figure 2A). Significantly differentially expressed genes (DEGs) were identified in the lungs of MWCNT-treated rats at each time point. A total of 1216 (854 upregulated and 362 downregulated), 1385 (875 upregulated and 510 downregulated), and 2213 (1142 upregulated and 1071 downregulated) genes were detected as DEGs at Year 0.5, Year 1, and Year 2, respectively (a full list of identified DEGs can be found in Appendix A). Among these DEGs, 738 genes were common to all three time points, while 142, 208, and 900 genes were exclusively detected at Year 0.5, Year 1, and Year 2, respectively (Figure 2B). Hierarchical clustering of the expression data of all DEGs showed a marked difference between control groups and MWCNT-treated groups and relatively tight clustering of MWNCT-treated animals in the Year 2 group (Figure 2C).

The top 20 upregulated or downregulated DEGs at each time point are listed in Table 1. *Spp1* (secreted phosphoprotein 1), *Ankrd34c* (ankyrin repeat domain 34A), *Mmp7* (matrix metallopeptidase 7), *Srd5a2* (steroid 5 alpha-reductase 2), *Mt3* (metallothionein 3), *Mmp12* (matrix metallopeptidase 12), and *Lpo* (lactoperoxidase) were upregulated at all three time points. *Dlk1* (delta-like non-canonical Notch ligand 1) and *Myl2* (myosin light chain 2) were downregulated at all three time points.

### 3.2. Ingenuity Pathway Analysis (IPA)

Figure 3 shows the Disease and Biofunction analysis of the DEGs in the lung tissue of MWNT-7-exposed rats. Disease and biofunctions in the cancer category were markedly altered at all time points, with a notably large number of DEGs at Year 2. Growth and development-related biological processes tended to be perturbed at Year 0.5–Year 1, with perturbation decreasing or absent at Year 2; this may reflect increased repair of lung tissue in the MWNT-7-exposed rats. Lipid metabolism, connective tissue disorders, cardiovascular disease, and neurological disease pathways were perturbed in Year 2. This could be related to the effect that chronic exposure to MWNT-7 had on the health of the rats exposed to these fibers. It is reasonable that if MWNT-7 affected lung function, chronically exposed rats could experience adverse health effects not related to cancer, an important endpoint that merits further investigation.

Significantly perturbed canonical pathways in MWNT-7-treated rats compared with the controls at each time point are shown in Figure 4. As in the Venn diagram of the DEGs (Figure 2B), many perturbed pathways were common to all three time points, and a relatively high number of pathways (78) were exclusively perturbed at Year 2 (Figure 4A). Figure 4B–E shows the top canonical pathways that are common to all three time points and the pathways that were perturbed exclusively at one of the three time points (all significant canonical pathways are shown in Appendix A). Throughout the experimental period, pathways associated with immune responses, especially macrophage activities, were perturbed (Figure 4B): *Phagosome Formation*, *Acute Phase Response Signaling* (Figure 5A), *Toll-like Receptor Signaling*, *Antioxidant Action of Vitamin C* (Figure 5B), *Complement System*, *Agranulocyte Adhesion and Diapedesis*, and *G-Protein Coupled Receptor Signaling*. Other examples of pathways that were perturbed at all three time points include Tumor *Microenvironment Pathway* (Figure 5C), *Hepatic Fibrosis/Hepatic Stellate Cell Activation*, and *Axonal Guidance Signaling* (Figure 4B). In these top-ranked pathways, individual data sets were well-clustered according to the time point (Year 0.5, Year 1, and Year 2) in the heatmaps of DEG expression (Figure 5A–C). In addition, time-dependent alterations of gene expression levels were generally observed.

At Year 0.5, 16 canonical pathways were exclusively perturbed, including activation of *Cyclins and Cell Cycle Regulation* and *Mitotic Roles of Polo-Like Kinases* (Figure 4C and Appendix A). At Year 1, 15 pathways, including metabolic and biogenesis-related pathways such as *Gluconeogenesis I* and *Chondroitin Sulfate Biosynthesis*, were perturbed (Figure 4D; Appendix A). At Year 2, a large number of pathways related to cancer and neural diseases were perturbed (Figure 4E), consistent with the results of Diseases and Biofunctions analysis (Figure 3). IPA identified several pathways implicated in cell proliferation and cancer biological processes, such as *HOTAIR Regulatory Pathway*, *MSP-RON Signaling In Cancer Cells Pathway* (Figure 6A), *Basal Cell Carcinoma Signaling*, *FAK Signaling*, *14-3-3-mediated Signaling*, and *ERBB Signaling* (Figure 6B). The most highly inhibited pathway was *Apelin Cardiomyocyte Signaling Pathway* (Figure 4E and Figure 6C). Several axonal guidance signaling pathways were also perturbed in Year 2: *Axonal Guidance Signaling* (Figure 4B), *Ephrin B Signaling* (Figure 4E), *Netrin Signaling* (Figure 4E), *Ephrin Receptor Signaling* (Figure 6D), and *Notch Signaling* (Appendix A). These pathways are associated with the regulation of cell growth and movement. In addition, similar to the results of Year 1 (Appendix A), metabolic and biosynthesis pathways such as *Phospholipase* (Figure 4E), *FXR/RXR Activation*, *LXR/RXR Activation*, *Cysteine Biosynthesis/Homocysteine Degradation*, *PPARα/RXRα Activation*, and *Pyrimidine Deoxyribonucleotides De Novo Biosynthesis I* (Appendix A) were perturbed.

Fourteen of the 20 upstream regulators identified by Upstream Regulator analysis were common to all three time points: *lipopolysaccharide (LPS)*, *CSF2*, *immunoglobulin*, *beta-estradiol*, *IFNG*, *TNF*, *IL10*, *IL6*, *TGFB1*, *AGT*, *IL1B*, *dexamethasone*, *IL13*, and *IL4* (Appendix A). All eight cytokines identified by Upstream Regulator analysis at Year 0.5 were also identified by Upstream Regulator analysis at Year 1 and Year 2. They are involved in the regulation of critical intracellular pathways, including STAT signaling and activation of the transcription factors *NFKB1* and *CEBPB* (Appendix A).

In the Regulator Effects analysis, *MYD88* showed the highest consistency score at Year 2 (Appendix A). MYD88 is an adaptor molecule that is essential in Toll-like receptor and interleukin-1 receptor signaling. One of its primary effects is the induction of macrophage chemotaxis in response to Toll-like receptor signaling (Appendix A). The Regulator Effects list at Year 2 also contains three regulators of lung tumorigenesis: *MALT1*, *AREG*, and *PLAUR* (Appendix A). Notably, the downregulation of *NOTCH4* is linked to breast and pancreatic cancer (Appendix A), suggesting cancer-associated roles of axon guidance molecules in lung carcinogenesis.

To integrate the findings from the Core Analysis of IPA, we utilized the Graphical Summary feature with a hierarchical representation of the key regulators and key biological functions found at each time point (Appendix A). For Year 0.5, the major activated regulator was *IL1B*, with links to several immune-signaling molecules and Toll-like receptor signaling (Appendix A). These key molecules are associated with the inflammatory response (releases of *IL1A* and *TNF*) as well as macrophage activities such as phagocytosis and chemotaxes. Inhibition of *SIGIRR*, a negative regulator of the Toll-like receptor signaling, and *ZFP36* (ZFP36 ring finger protein/tristetraprolin), an anti-inflammatory modulator in murine models of systemic inflammatory diseases, were also involved in this network. At Year 1, *SPP1* was identified as the major regulator. The *SPP1* network contains several immune-related proteins as well as cancer-related proteins (e.g., *Neoplasia of cells*, *CCND1*, *Thoracic neoplasm* and *Metastatic solid tumor*) (Appendix A). The immune function-associated proteins *CSF2*, *IKBKB*, and *IL1A* were common to the Year 0.5 and Year 1 networks. At Year 2, *SPP1* was again identified as a main regulator, but neoplastic outcomes were more frequently represented compared with the SPP1 network of Year 1. A set of genes involved in epithelial cell growth (*ERBB2*, *EGF*, and *FGF2*) and *Plasminogen (PLG)* were connected to *Cancer* and *Lung cancer* in the Year 2 SPP1 network (Appendix A).

### 3.3. qRT-PCR

To validate RNA-seq data, we measured the expression of 22 selected genes by qRT-PCR (Figure 7). The results of gene expression determined by qRT-PCR were consistent with the RNA-seq data (see Figure 5 and Figure 6; a full list is shown in Appendix A). Three of the top 20 DEGs, *Spp1*, *Mt3*, and *Mmp12*, were examined. Expression levels of all three genes were substantially increased compared to time-matched controls, and *Mt3* and *Mmp12* showed clear time-dependent increases. Three inflammation-related genes, *Ccl2* (C-C motif chemokine ligand 2), *Il6* (interleukin 6), and *Zfp36*, were examined. Expression levels of the pro-inflammatory genes *Ccl2* and *Il6* were upregulated while expression of the anti-inflammatory gene *Zfp36* was downregulated. A protease regulating plasmin production, *Plau* (plasminogen activator, urokinase), and its receptor, *Plaur* (plasminogen activator, urokinase receptor), were also significantly elevated. *Plau* was significantly upregulated only at Year 2. Several ERBB signaling molecules, including receptors *Egfr* (epidermal growth factor receptor), *Erbb2* (Erb-B2 receptor tyrosine kinase 2), *Erbb3*, and *Erbb4*, and their ligands, *Egf* (epidermal growth factor), *Areg* (amphilegurin), *Ereg* (epilegurin), *Btc* (betacellulin), and *Nrg1* (neuregulin 1), were measured. Expression of these genes was in agreement with the RNA-seq data. The expression of three genes involved in negatively regulated canonical pathways, *Aplnr* (apelin receptor/APJ) and two axon guidance proteins, *Ephb4* (EPH receptor B4) and *Unc5d* (unc-5 Netrin receptor D), was examined. Expression of all three genes was significantly decreased at Year 2. Finally, expression of two known as tumor suppressor genes, *Cav1* (caveolin 1) and *Wif1* (wingless-type inhibitory factor-1) was examined. In the control group, the highest expression level of *Cav1* was at Year 2, and at Year 2, *Cav1* was significantly decreased in the MWCNT-treated group. *Wif1* expression was downregulated at all three time points.

### 3.4. Immunohistochemistry

We further investigated the spatial localization of key molecules at Year 0.5 and Year 2 by immunohistochemistry (Figure 8 and Figure 9). As shown by H&E staining, macrophage aggregations, fibrotic and granulomatous changes, and reactive hyperplasias of type II pneumocytes were the major findings at Year 0.5 (Figure 8), while preneoplastic lesions, i.e., atypical hyperplasias, mainly composed of proliferative cells possibly derived from bronchiolar cells (Figure 9 center column) or alveolar cells (Figure 9 right column) were frequently observed at Year 2 (also see Appendix A). There was strong staining of SPP1 in alveolar macrophages, especially in large foamy or fractured cells at Year 2. The SPP1 protein was also induced in epithelial cells in the two types of atypical hyperplasias, the one that appeared to be derived from bronchiolar cells and the one that appeared to be derived from alveolar cells. HMOX1 (heme oxygenase 1) was detected in alveolar macrophages at Year 0.5, and strong staining was noted in macrophages as well as hyperplastic epithelial cells, possibly derived from the bronchiole, at Year 2. LPO staining resembled that of HMOX1, but strong staining was observed in both types of preneoplastic lesions. MMP12 was highly localized to normal bronchial epithelial cells, but mild staining was also detected in MWCNT-laden macrophages, alveolar walls with granulomatous and fibrotic changes, and preneoplastic lesions. Three ERBB signaling-related proteins, EGFR, ERBB2, and AREG, exhibited a similar pattern. The above proteins localized to macrophages and bronchiolar and alveolar cells, and were markedly stained in preneoplastic lesions. ANKS1B (ankyrin repeat and sterile alpha motif domain containing 1B) was mainly detected in bronchiolar cells but staining was also observed in active macrophages and proliferative epithelial cells.

## 4. Discussion

MWNT-7 is one of the most important reference MWCNTs due to an abundance of positive toxicological data, including lung carcinogenicity by inhalation in rats [14], lung carcinogenicity by intratracheal instillation in rats [29], carcinogenicity of the peritoneum and pleura in rats by intraperitoneal injection [9,33] and intratracheal spraying/instillation [29,34], and genotoxicity by in vitro and in vivo tests [35,36,37]. In the present study, we first performed a profiling of transcripts sourced from rats from a 2-year study in which a significantly high incidence of lung tumors was observed in the MWNT-7-treated group [29]. The tissues were from control rats and from non-tumor lung tissue of rats exposed to MWNT-7 for 0.5, 1, and 2 years. A major biological event at all three time points was macrophage-elicited inflammatory responses, suggesting inflammation-based carcinogenicity. The top-ranked perturbed pathways identified in the present study were consistent with those of previous analyses of CNT-exposed rat and mouse lungs, especially with regard to inflammation (Appendix A) [16,17,18,22,24,38,39,40,41,42]. In the present study, we also found that several of the perturbed biofunctions increased in a time-dependent manner (see Figure 3). In addition, there was a tendency for altered gene expression levels to become greater over time (see Figure 5 and Table 1). Expression of Hmox1 (heme oxygenase 1), a well-known marker of ROS generation, also increased from Year 1 to Year 2. This contrasts with previous time-course studies in which inflammation-related pathways or gene ontology terms were significantly enriched 3–7 days after single exposures and markedly declined during the recovery period [16,17]. Our transcriptomic analysis suggests that inflammation-related ROS generation is likely a cause of lung carcinogenicity, but a persistent inflammatory stimulus may be needed to induce lung tumorigenesis. Our data showed no clear switch from the inflammatory response to other major biological events. Rather, the inflammation response lasted and escalated through the experiment, and additional events occurred at later phases (Figure 10). Several in vivo and in vitro studies suggested that CNTs activate the NFB-mediated inflammasome and produce IL-1, leading to the generation of ROS and reactive nitrogen species in immune cells and surrounding alveolar and bronchiolar cells [3,43,44]. In addition, genotoxicity studies have indicated that some types of CNT, including MWNT-7, are genotoxic and have reported induction of double-strand breaks and positive results using the comet assay [4]. However, MWCNT genotoxicity in vivo arises through a secondary mechanism [4,45]. Totsuka and colleagues demonstrated that MWNT-7 induced ROS-induced mutation signatures such as G-C transversion in mice [35,45]. ROS generation by macrophages could result in these mutations being fixed in the cell genome, especially during chronic inflammation and the resulting rounds of tissue damage and tissue repair. Consequently, doses of MWCNT that result in chronic inflammation, which our analysis indicated occurred in the present study, could result in mutations of key driver genes over time. In a human sequential lung cancer development scheme, early events such as loss of heterogeneity, microsatellite alteration, and small telomeric deletions precede morphological abnormalities (e.g., hyperplasia, dysplasia, and carcinoma in situ) and mutations of key genes such as *Egfr*, *Plau*, *p53*, and *Kras* [46,47]. In a rat silica-induced lung cancer model, the DNA damage response such as H2AX was evident in early preneoplasia [48]. Therefore, future studies should examine the early genetic (and epigenetic) alterations in the lung during MWNT-7-induced lung cancer in rats and mice. DNA mutation signature analyses and adductome analysis will help to establish whether inflammation-related ROS actually does result in genomic damage during chronic MWCNT exposure [49,50,51].

In addition to the inflammation-related bioprocesses that were detected at Year 0.5, Year 1, and Year 2, IPA identified several time point-specific biological functions and pathways. For example, pathways associated with cell cycle and cyclin regulation were identified as being specific to Year 0.5. These pathways were, however, not necessarily exclusive to Year 0.5. The DEGs involved in these pathways were also upregulated or downregulated in samples from the other two time points, as shown in the heatmap in Appendix A (compare the M1 samples with the M2 and M3 samples). While the *p*-value of this pathway did not exceed the threshold *p* value (−log (*p*-value) > 1.3), activation of this pathway at Year 1 and Year 2 is likely (Figure 10). 

At Year 1, activation of some metabolic and biogenesis-related pathways was noted; however, activation of these pathways seems not to be specific to Year 1, but rather to be maintained until Year 2 (phospholipase and some nuclear receptor signaling pathways) (Figure 10). Recently, impairment of pulmonary lipid homeostasis has been addressed in studies of the toxicities of electronic cigarettes and nanomaterials, including CNTs [22,52,53,54,55]. It was found that this could profoundly affect surfactant metabolism, lamellar body biogenesis in type II pneumocytes, and the plasticity and functions of alveolar macrophages.

At Year 2, several interesting canonical pathways were identified (Figure 4 and Figure 10), many of which are not included in previous toxicogenomic reports analyzing CNTs (Appendix A) [16,17,18,22,24,38,39,40,41,42]. *ERBB Signaling Pathway* was significantly perturbed, but IPA analysis could not determine whether this pathway was upregulated or downregulated (z-score: −0.9). Considering the wide variety of combinations of receptors and their ligands and crosstalk with other intracellular signaling pathways such as G-protein-coupled receptors and insulin-like growth factor receptors, the results extracted from IPA need to be more closely scrutinized. EGFR and ERBB2/Her2 are well-known oncogenic genes in human lung adenocarcinoma [56,57,58]. ERBB2 does not have a ligand binding domain of its own; rather, it binds to ligand-bound EGF receptor family members, stabilizing ligand binding and enhancing downstream signaling. Detailed histological analyses revealed stepwise increases in the immunohistochemical signals from normal bronchial mucosa to epithelial hyperplasia to cancer [46,56,59]. Prominent protein localizations in the preneoplastic foci in the present study are suggestive of a contribution of the two receptors to CNT-induced carcinogenesis. Among 12 ligands for ERBB families, those for ERBB3 and ERBB4 (e.g., *Nrg1*, *Nrg2*, and *Nrg3*) showed minimal or no significant alterations of gene expression in RNA-seq, while some ligands that bind mainly to EGFR (e.g., *Btc*, *Ereg*, and *Areg*) exhibited significant upregulation in RNA-seq and qRT-PCR validation, suggesting activation of EGFR at Year 2. However, levels of transcripts of ERBB family genes were not increased in qRT-PCR. The expressions of *Egfr* and *Erbb3* exhibited no or limited changes, while *Erbb2* and *Erbb4* decreased. Downregulation of the *Erbb4* transcript was notable in the results of qRT-PCR. ERBB4 is the only member of the ERBB family with a potential function as a tumor suppressor gene, and loss of the gene copy number of ERBB4 was found in approximately 20% of human lung squamous carcinoma and adenocarcinoma patients [60].

*MSP-RON Signaling In Cancer Cells Pathway* is another pathway that was exclusively perturbed at Year 2. Macrophage-stimulating protein (MSP) is a plasminogen-related growth factor and binds to d’Origine Nantais (RON), a receptor tyrosine kinase belonging to the MET proto-oncogene family. The MSP-RON signal plays a role in inflammation and innate immunity, modulating macrophage features in acute and chronic inflammation and tumor-immune escape [61]. Additionally, MSP-RON was shown to be involved in cell proliferation, migration, and invasion in human pulmonary cancer [62,63]. Notably, the MSP-RON axis upregulates gene expression of PLAUR [64]. PLAU (plasminogen activator, urokinase) and its receptor PLAUR (plasminogen activator, urokinase receptor) influence many normal and pathological processes related to cell-surface plasminogen activation and localized degradation of the extracellular matrix (ECM) [65]. Thus, PLAUR is also involved in the *Tumor Microenvironment* pathway. The MSP-RON pathway and PLAU-PLAUR system possibly have roles in both regulating macrophages and neoplastic changes of alveolar and bronchiolar cells in chronic inflammation caused by MWNT-7.

Guidance molecules are critical players in the nervous system to control axon outgrowth and direction, but they are also widely expressed outside the nervous system, where they control cell migration, tissue development, and the establishment of the vascular network. They are also involved in lung cancer development and metastasis [66]. One such signaling pathway is the Slit-Robo pathway; however, Slit-Robo signaling, which correlates with altered cell motility, can have opposing effects in different cancers [67]. These opposing effects are attributed to the variety of ligands and receptors that are involved in cell motility in different environments. In the lung, Slit/Robo suppresses tumor formation [67], and Robo1^−/−^ and Robo1^+/−^ mice have a high incidence of alveolar hyperplasia and exhibit increased susceptibility to lung cancer [68,69]. Notably, several axonal guidance signaling pathways were perturbed at Year 2: *Axonal Guidance Signaling* was perturbed at all three time points, and *Ephrin B Signaling*, *Netrin Signaling*, *Ephrin Receptor Signaling*, and *Notch Signaling* were all downregulated at Year 2. In addition, another guidance molecule, Notch4, was identified as an upstream regulator (Appendix A). These results suggest that genes associated with axonal guidance play an important role in the response to MWCNT exposure in the lung. In rats, reactive hyperplasia of alveolar cells occurs as a response to damage to the alveolar epithelium by inhalation of insoluble particles or fibers, while atypical hyperplasia arises as a “preneoplastic” lesion, leading to lung adenoma or adenocarcinoma. Similar to the findings in 2-year inhalation studies of indium phosphide, indium-tin oxide, and MWNT-7 [14,70,71], we found atypical hyperplasia in the lungs of MWCNT-treated rats at Year 2 in the rats used for evaluation of carcinogenicity [29]. Downregulation of guidance molecule pathways could link the structural alteration from the reactive (reversible) response to MWCNT to preneoplastic lesions in the lung.

Another interesting pathway to emerge from this study is the *Apelin Cardiomyocyte Signaling Pathway*, identified as the most negatively regulated pathway in Year 2. Apelin is an endogenous ligand for the APJ receptor (Apelin receptor; APLNR). The apelin-APJ system plays various roles in the physiology and pathophysiology of many organs, including the regulation of cardiac contractility, angiogenesis, cell proliferation, and apoptosis [72]. Recently, the apelin-APJ axis has been shown to alleviate fibrosis in several organs by regulating PI3K/Akt signaling and TGF-β signaling [73,74]. In an LPS-induced pulmonary fibrosis model, the apelin-APJ axis was found to regulate fibrosis through TGF-β1-mediated endothelium-mesenchymal transformation (EndMT). In the present study, severe pulmonary fibrosis was histologically evident, especially at Year 2 [29], and the *Hepatic Fibrosis/Hepatic Stellate Cell Activation pathway*, *Hepatic Fibrosis Signaling Pathway*, and *Pulmonary Fibrosis Idiopathic Signaling Pathway* were also perturbed, especially at Year 2 (compare Appendix A with Appendix A). Similar to the LPS model, repeated MWCNT exposures likely inhibited the apelin-APJ axis, which would exacerbate fibrosis by promoting EndMT in the lung parenchyma. In humans, fibrosis is not only a severe adverse event in the lung but also a risk factor for lung cancer development, probably due to the shared cellular and molecular processes driving the progression of both pathologies [75]. The prominent fibrosis was observed in association with atypical hyperplasias in rat 2-year carcinogenicity studies for MWNT-7, indium phosphide, and indium-tin oxide [14,70,71]. Fibrosis is one of the common outcomes of CNT exposures in rodent studies, even with short-term experiments, but the relationship between CNT-induced fibrosis and lung cancer development remains unclear, warranting further investigation.

We also characterized temporal expression patterns of some DEGs. *Spp1* was shown to be a major regulator in our data set (Appendix A). SPP1/osteopontin was first identified as a bone matrix protein regulating bone remodeling, but its diverse biological roles are becoming increasingly recognized, especially in the regulation of the immune response to chronic inflammation and tumorigenesis [76,77]. Giopanou et al. examined *Spp1* function in urethane-induced murine lung adenocarcinoma development using *Spp*^−/−^ and Cre-loxP-*Kras^G12D^* mice. Epithelial and macrophage-secreted SPP1 activated tumor-associated inflammation, and interestingly, the epithelial SPP1 promoted early tumorigenesis by fostering the mutant *Kra^sG12D^*-expressing cells [78]. Similar to their immunostaining results, we found localization of the SPP1 protein not only in macrophages and normal bronchiolar epithelium but also in preneoplastic epithelial cells. Accordingly, rat *Spp1* may have roles in regulating chronic inflammation and the onset of lung carcinogenesis induced by MWCNT exposure.

*MMP12* plays a role in degrading ECM and contributing to remodeling damaged tissues. Some researchers have reported high levels of *MMP12* induction in macrophages and surrounding tissues as a response to carbon-based nanomaterials [17,79,80]. In the present study, marked upregulation and time-dependent increases at both transcript and protein levels were observed, suggesting the participation of *Mmp12* in the worsening of histological findings such as fibrosis, granulation, and proliferation of epithelial cells.

*Mt3* and *Lpo* were also found to be markedly upregulated at all three time points and upregulation increased from Year 0.5 to Year 2. MT3 is a zinc-binding protein that can bind a variety of heavy metal ions as well as oxygen and nitrogen radicals, and recently it has been recognized as a multifunctional player in redox, apoptosis, lysosomal biogenesis, and carcinogenesis [81,82]. LPO is a type of peroxidase secreted by goblet cells in the epithelial lining of the respiratory tract. In the presence of hydrogen peroxide, LPO oxidizes thiocyanate to generate hypothiocyanite, a potent antimicrobial [83]. These genes are possibly related to inflammation-induced carcinogenesis and may be biomarker candidates for chronic MWCNT toxicities.

ANKS1B is a multi-domain protein that has a role in the pathogenesis of Alzheimer’s disease. A systemic search for SNPs in apoptotic-pathway genes revealed a relationship between the SNPs of *ANKS1B* and lung cancer risk in humans, but their roles in lung cancer development are largely unknown [84]. The circular RNA of *Anks1b* promotes metastasis of triple-negative breast carcinoma by modulating epithelial-to-mesenchymal transition [85]. In the present study, gene expression was markedly elevated at Year 1, and at Year 2 it was the fifth most upregulated gene. Although the specific role of circ-Anks1b in rodent lungs has not been studied, the strong detection of the protein in epithelial cells at Year 2 is suggestive of a role for this gene in the onset of lung cancer development in rats.

Finally, key driver genes or potential biomarkers of human lung cancer were altered in the present study (Appendix A) [86,87,88,89,90,91,92]. Besides markedly upregulated genes, such as Spp1, Mmp12, and Mt3, we focused on two downregulated genes. *Cav1* is suggested to act as a tumor suppressor in various cancers, including lung adenocarcinoma, by regulating cell proliferation and cell death [93,94]. *Wif1*, an important regulator in the WNT pathway, is also a well-known tumor suppressor gene in lung cancer [95,96]. Both *Cav1* and *Wif1* were appreciably downregulated at Year 2.

The present study has two major limitations. To evaluate preneoplastic status in this study, we excluded tumor nodules; however, the frozen samples could have included various focal lesions such as reactive alveolar hyperplasias, atypical hyperplasias, and early adenomatous lesions. In addition, perturbation of proinflammatory and profibrogenic genes in immune cells and interstitial cells could have masked alterations in preneoplastic epithelial cells. Analyses with high spatial resolution using laser microdissection or approaches such as spatial transcriptomics and photo-isolation chemistry are needed [97,98]. The other limitation is that we analyzed only one type of MWCNT and used animals exposed to only one dosage level of this MWCNT. Previous toxicogenomic analyses for CNTs have clearly demonstrated dose-dependent results [16,38], and some comparative analyses have discovered some physicochemical feature-dependent gene expression patterns [22,23]. In particular, if we had prepared another negative control group (dosed with a “non-carcinogenic fiber”), it would have provided a better comparison. In a rat peritoneal injection model, a comparison between rats injected with a “carcinogenic fiber” (MWNT-7) and those injected with a thin, tangled, “non-carcinogenic fiber” revealed a transcriptomic signature of the peritoneal macrophages contributing to mesothelioma development [99]. However, regarding lung carcinogenicity, relevant non-carcinogenic fibers have not been identified despite intensive 2-year studies evaluating the carcinogenicities of various CNTs administered by an intratracheal exposure protocol termed “intra-Tracheal Intra-Pulmonary Spraying (TIPS)” (established by Tsuda et al.) [25,26]. In our recent intratracheal spraying study (manuscript in preparation), a mill-ground MWNT-7 (shortened MWNT-7) induced only weak inflammation and resulted in no lung tumor development after 2 years despite administering a total dose of 1.2 mg/rat, a dose that was comparable to the studies by Tsuda et al. [25,26]. The shortened MWNT-7 may become a negative control when evaluating the lung carcinogenicity of CNTs in rats.

While the entire molecular mechanism of CNT-induced pulmonary carcinogenesis is still far from being completely established, longitudinal assessment of the transcriptome before and during carcinogenesis is a fundamental step toward effective mechanistic analyses in animal experiments and in vitro studies. It is hoped that further studies will evaluate whether the molecules and pathways found in this study are indeed involved in carcinogenesis and are applicable to toxicological biomarkers using knockdown or inhibition techniques. As mentioned above, omics studies with other experimental designs or using other types of CNTs will give new insights into the molecular mechanism of CNT-induced lung carcinogenesis. Promising protocols as alternatives to 2-year inhalation tests (such as TIPS) may render omics applications to carcinogenic endpoints relatively easy. 

Regarding the relevance of biomarkers for exposure to CNTs in workplaces, the results of Year 0.5 seem similar to those of microarray analysis of mRNA and miRNA in the blood of workers in the CNT factory, in which many inflammatory genes were altered and a network with a hub gene, Cyclin D1, suggestive of the perturbation of the cell cycle, was predicted by IPA [21]. Thus, damage by CNT and regenerative change might occur in the respiratory system of the human lungs. Although there are some difficulties in extrapolating the results of rat lung carcinogenicity tests to humans in risk assessment [100,101,102,103,104], from a no-observed-adverse-effect level (NOAEL) of rat lung carcinogenicity of MWNT-7 (0.02 mg/m^3^ [14]), an occupational exposure limit (OEL) was calculated as 0.15 μg/m^3^ [105,106]. This OEL value is lower than the recommended exposure limit (REL; 1 μg/m^3^) by the National Institute of Occupational Safety and Health (NIOSH) or other proposed OELs [107], and notably, occupational exposure research showed that measured values sometimes exceeded the NIOSH REL [21,108,109]. These facts highlight the need for comprehensive analyses for biomarker candidates using liquid biopsy (such as [21,110]). In this regard, we have started a new comprehensive analysis of rat sera obtained from the same 2-year experiment as this study. The results of such an analysis will be more comparable to those of occupational exposure surveys using liquid biopsy, and integration of the readouts of the transcriptome from serum and lungs will give promising candidates for mechanistic biomarkers.

## 5. Conclusions

Our time-course transcriptomic profiling of rat lungs exposed to MWNT-7 for 2 years highlighted persistent inflammation and ROS generation associated with macrophage activities. These responses to MWNT-7 exposure were evident from the earliest time point examined, 0.5 years, to the 2-year time point, and pathways associated with inflammation, ROS generation, and tissue damage demonstrated a high level of concordance with perturbed pathways identified in other omics studies with relatively short-term exposure to CNTs. Continued exposure to MWNT-7 led to alterations in pathways associated with metabolic and biogenesis around 1 year after the beginning of the experiment and caused perturbation of several cancer-related signaling and guidance molecule signaling pathways at experimental termination, suggesting the emerging of neoplastic changes in the pulmonary epithelium. Our findings will increase our understanding of the molecular mechanism of CNT-induced lung carcinogenicity and can serve as a benchmark for comparing the molecular signatures of chronic toxicity of other CNTs.

## Figures and Tables

**Figure 1 nanomaterials-13-02105-f001:**
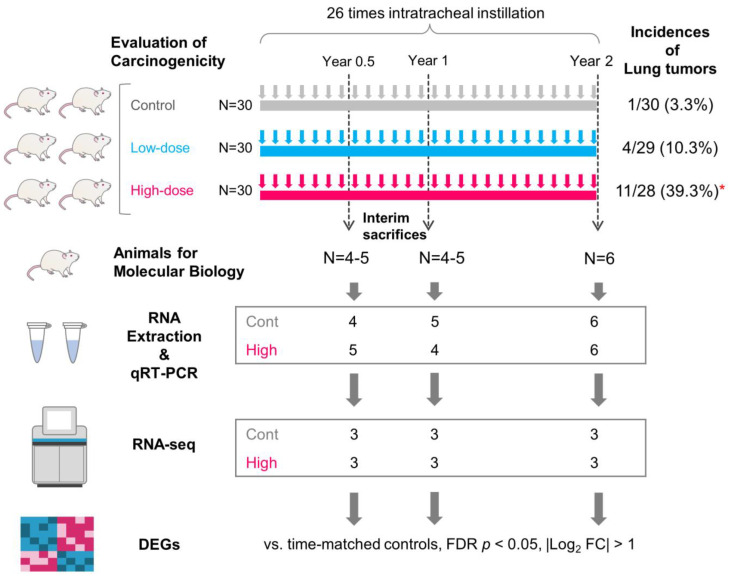
Experimental design. Schematic overview of MWCNT exposure, lung harvest, and gene expression analysis. The main 2-year experiment was previously performed for the evaluation of carcinogenicity of MWNT-7 (Hojo et al., 2022) [29]. Tumor incidences are shown on the upper right side. Asterisk: significant difference from the control group (by Fisher’s exact test). RNA was extracted from 4–5 animals at interim sacrifices (Year 0.5 and Year 1). In Year 2, RNA samples were obtained from six animals. Among these samples, 3 samples per time point per group were used for RNA-seq. All samples were used for qRT-PCR.

**Figure 2 nanomaterials-13-02105-f002:**
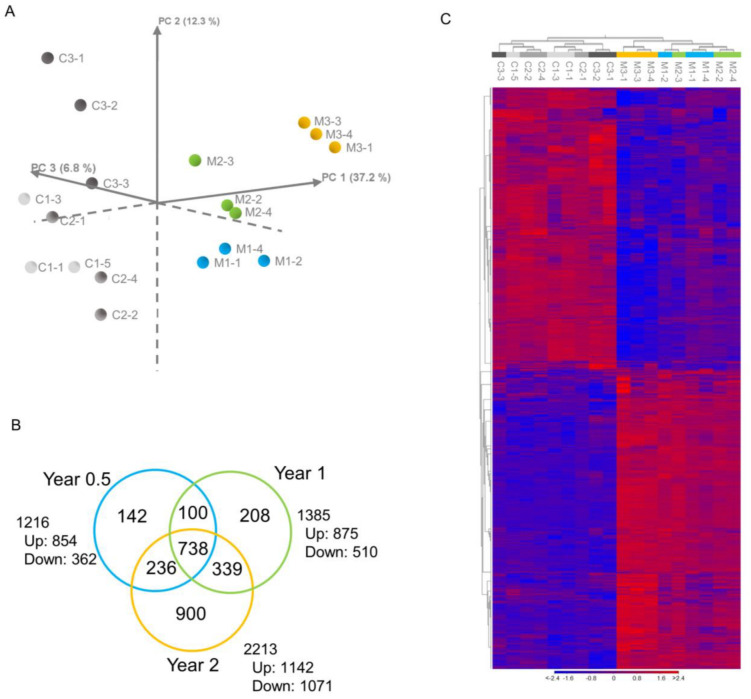
Overview of the gene expression analysis. (**A**) Principal component analysis (PCA) illustrating a marked difference in gene expression between the control and MWCNT-treated groups and similarities within Year 0.5, Year 1, and Year 2 of MWCNT-treated groups. (**B**) Venn diagram showing the distribution of DEGs between MWCNT-exposed lungs and control lungs at Year 0.5, Year 1, and Year 2. Both upregulated and downregulated genes were included. (**C**) Hierarchical clustering analysis of all DEGs. Color bar indicates high-expressed (red) and low-expressed (blue) genes. N = 3 per time point. Animal numbers of the control group are as follows. C1-1, C1-3, C1-5: Year 0.5; C2-1, C2-2, C2-4: Year 1; and C3-1, C3-2, C3-3: Year 2. Animal numbers of the MWCNT-treated group are as follows: M1-1, M1-2, M1-4: Year 0.5; M2-2, M2-3, M2-4: Year 1; M3-1, M3-3, M3-4: Year 2.

**Figure 3 nanomaterials-13-02105-f003:**
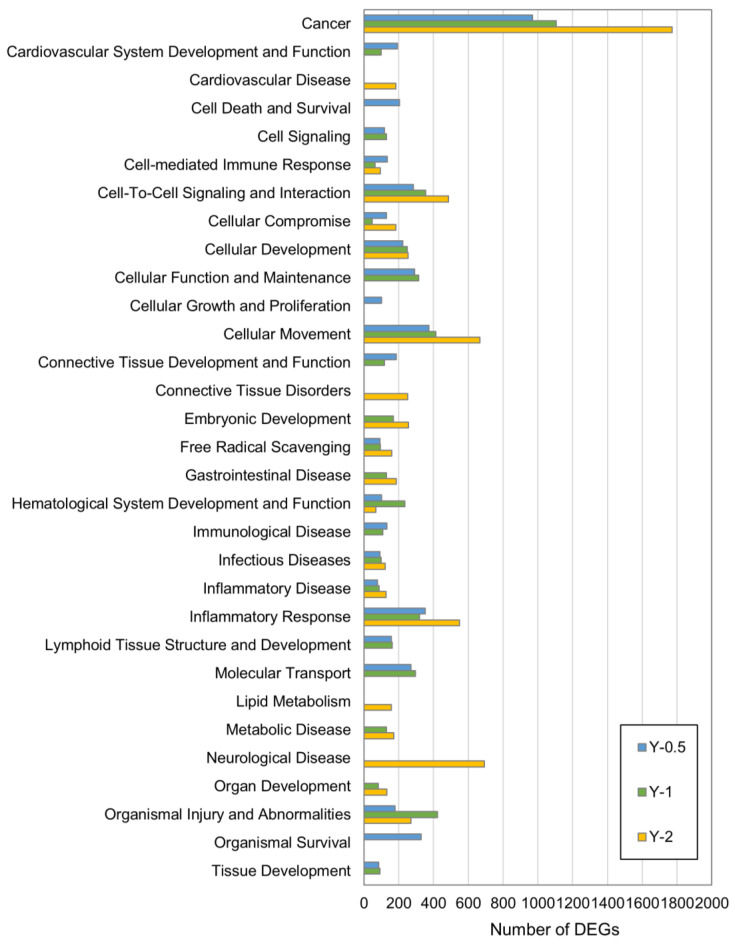
Disease and Biofunctions perturbed in the lungs of MWCNT-treated rats. The number of DEGs belonging to a set of IPA Disease and Biofunctions categories at each time point.

**Figure 4 nanomaterials-13-02105-f004:**
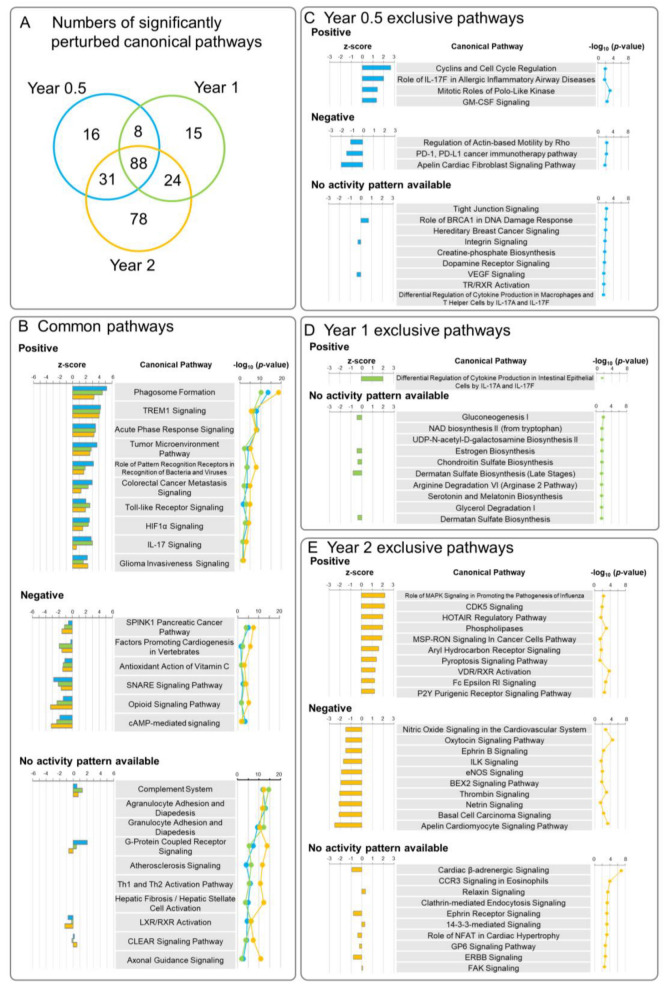
Canonical pathways perturbed in the lungs of MWCNT-treated rats. (**A**) Venn diagram showing the number of significantly perturbed canonical pathways in the MWCNT-treated group compared to time-matched controls at each time point. (**B**) Canonical pathways perturbed at all three time points. Among the 88 pathways, the top 10 (if applicable) pathways are arranged according to the mean −log_10_ (*p*-value) of the three time points with respect to positively regulated pathways, negatively regulated pathways, and pathways with no activity pattern available. (**C**) Canonical pathways exclusively perturbed at Year 0.5. All 16 pathways are listed. Positively and negatively regulated pathways are arranged according to the z-score. Pathways with no activity pattern available are arranged according to the −log_10_ (*p*-value). (**D**) Canonical pathways exclusively perturbed at Year 1. Among the 15 pathways, the top 10 (if applicable) pathways are listed as positively regulated pathways and pathways with no activity pattern available. No negatively regulated pathways were predicted. (**E**) Canonical pathways exclusively perturbed at Year 2. Among the 78 pathways, the top 10 pathways are listed for positively regulated pathways, negatively regulated pathways, and pathways with no activity pattern available. Positively and negatively regulated pathways are arranged according to the z-score. Pathways with no activity pattern available are arranged according to the −log_10_ (*p*-value). All perturbed canonical pathways are listed in Appendix A (Year 0.5), Appendix A (Year 1), and Appendix A (Year 2).

**Figure 5 nanomaterials-13-02105-f005:**
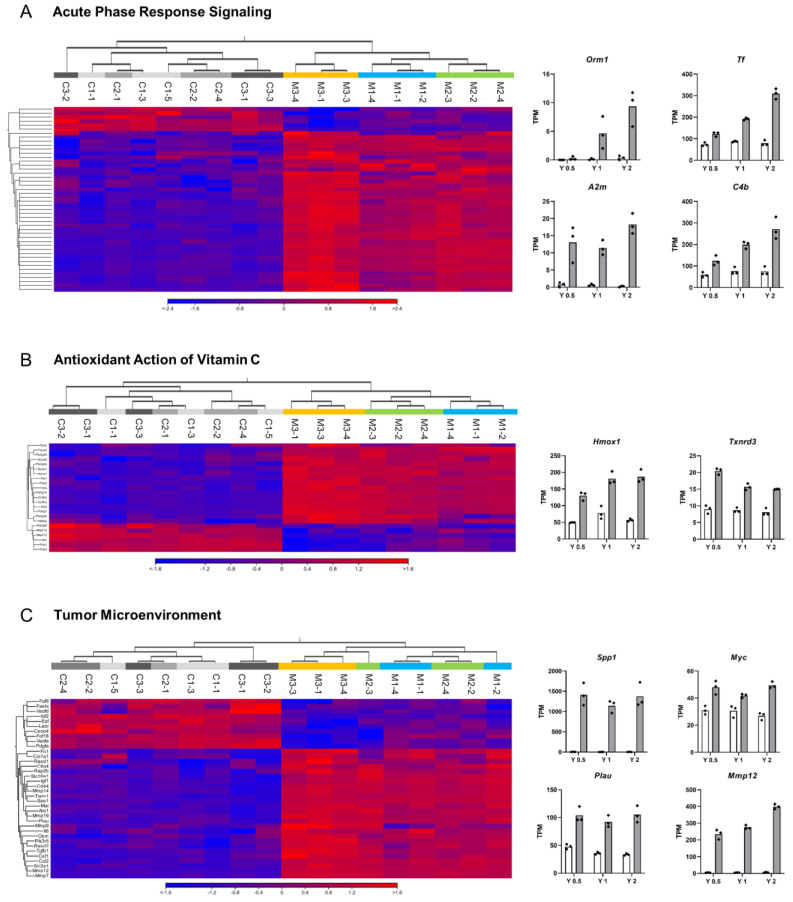
Gene expression changes in commonly perturbed pathways. Heatmaps resulting from hierarchical clustering of DEGs belonging to the pathway and bar graphs showing each RNA-seq data of key DEGs involved in the selected pathway. (**A**) Acute Phase Response Signaling. (**B**) Antioxidant Action of Vitamin C. (**C**) Tumor Microenvironment. Color bar indicates high-expressed (red) and low-expressed (blue) genes. N = 3 per time point. Bar graph shows the mean value of Transcripts per million (TPM) with individual data (dot). White bar graph: Control group. Gray bar graph: MWCNT-treated group. Animal numbers of the control group are as follows. C1-1, C1-3, C1-5: Year 0.5; C2-1, C2-2, C2-4: Year 1; and C3-1, C3-2, C3-3: Year 2. Animal numbers of the MWCNT-treated group are as follows: M1-1, M1-2, M1-4: Year 0.5; M2-2, M2-3, M2-4: Year 1; M3-1, M3-3, M3-4: Year 2.

**Figure 6 nanomaterials-13-02105-f006:**
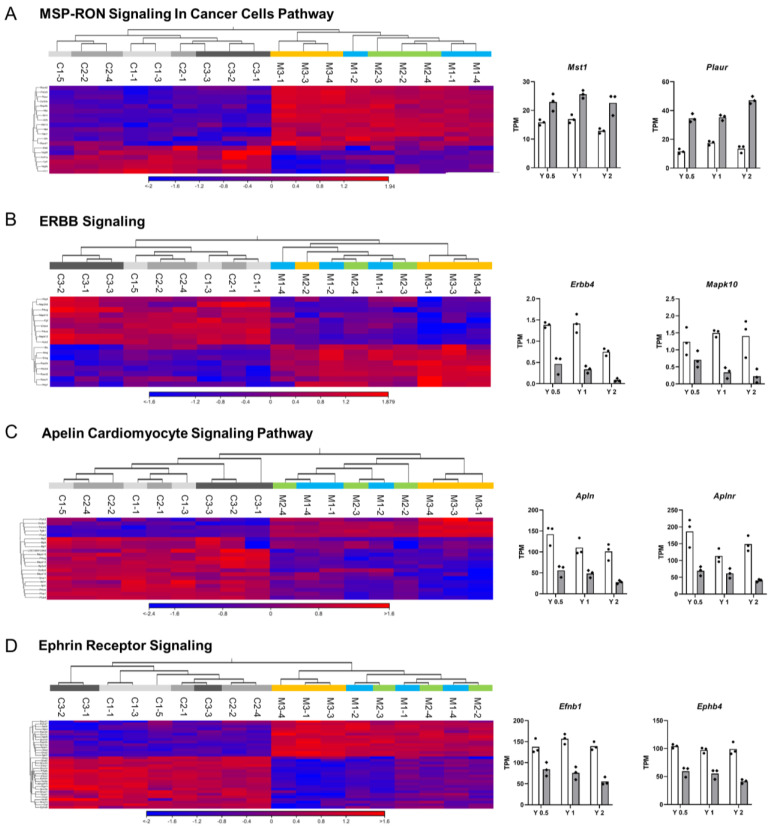
Gene expression changes in pathways perturbed at Year 2. Heatmaps resulting from hierarchical clustering of DEGs belonging to the pathway and bar graphs showing each RNA-seq data of key DEGs involved in the selected pathway. (**A**) MSP-RON Signaling In Cancer Cells Pathway. (**B**) ERBB Signaling. (**C**) Apelin Cardiomyocyte Signaling Pathway. (**D**) Ephrin Receptor Signaling. Color bar indicates high-expressed (red) and low-expressed (blue) genes. N = 3 per time point. Bar graph shows the mean value of Transcripts per million (TPM) with individual data (dot). White bar graph: Control group. Gray bar graph: MWCNT-treated group. Animal numbers of the control group are as follows. C1-1, C1-3, C1-5: Year 0.5; C2-1, C2-2, C2-4: Year 1; and C3-1, C3-2, C3-3: Year 2. Animal numbers of the MWCNT-treated group are as follows: M1-1, M1-2, M1-4: Year 0.5; M2-2, M2-3, M2-4: Year 1; M3-1, M3-3, M3-4: Year 2.

**Figure 7 nanomaterials-13-02105-f007:**
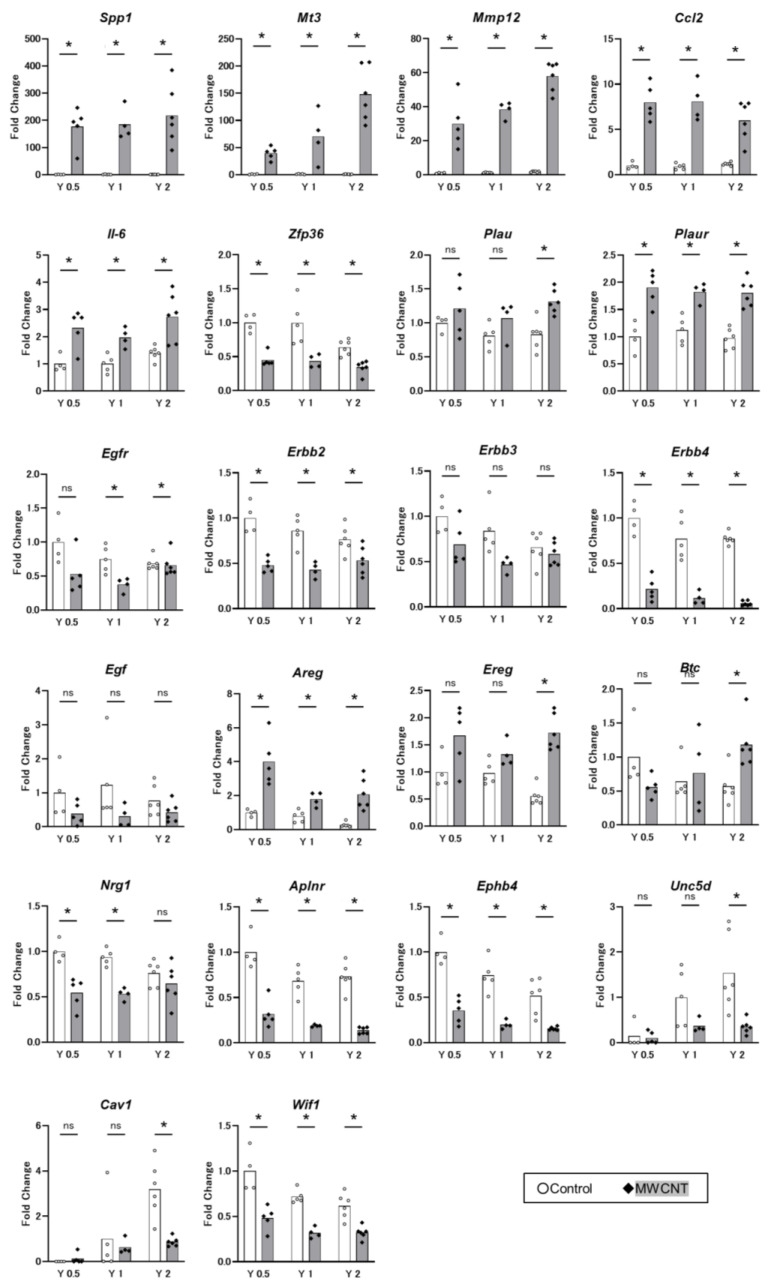
Gene expression levels of selected 22 genes measured by qRT-PCR. Gene symbols are shown at the top of the graph. Bars show mean and dots show individual data. White bar and white circle: Control group. Gray bar and black diamond: MWCNT-treated group. N = 4–6. Asterisk: differentially expressed compared with time-matched controls. *p* < 0.05. Each result is shown as a fold change relative to the control group at Year 0.5. For two genes, *Unc5d* and *Cav1*, data are shown as relatives to the control group at Year 1.

**Figure 8 nanomaterials-13-02105-f008:**
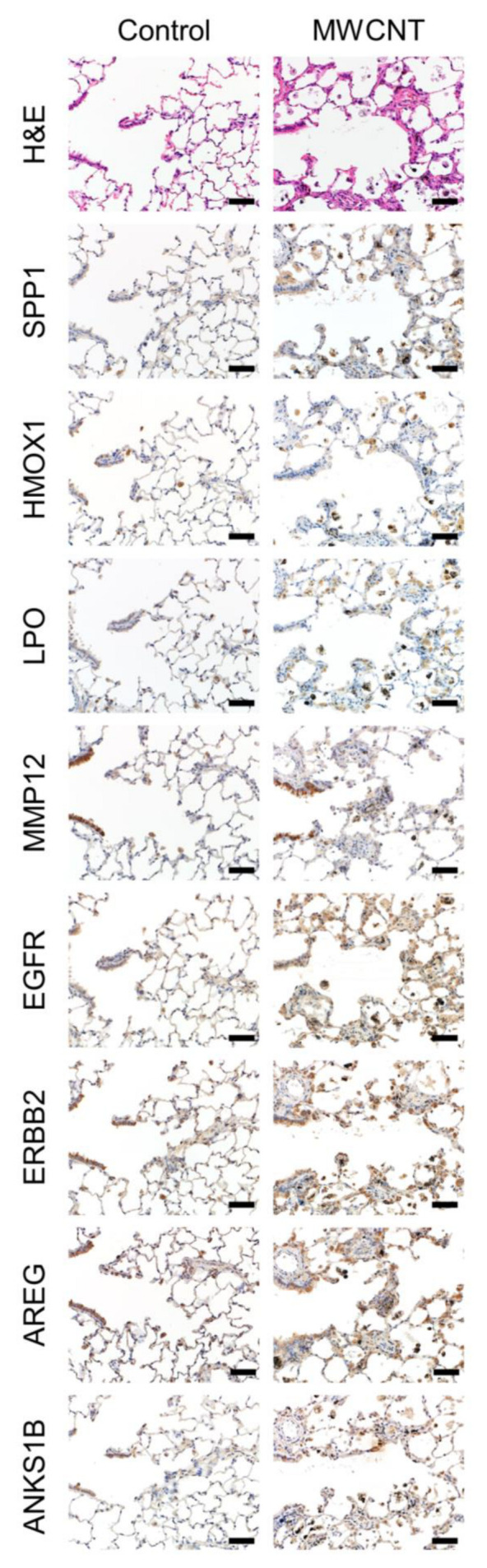
Immunohistochemistry for the selected eight proteins at Year 0.5. H&E (**top**) and immunohistochemical staining for the selected protein in serial sections of rat lungs of control (**left**) and MWCNT-treated (**right**) groups. Protein names are shown on the left side of the photograph. Scale bar: 50 µm.

**Figure 9 nanomaterials-13-02105-f009:**
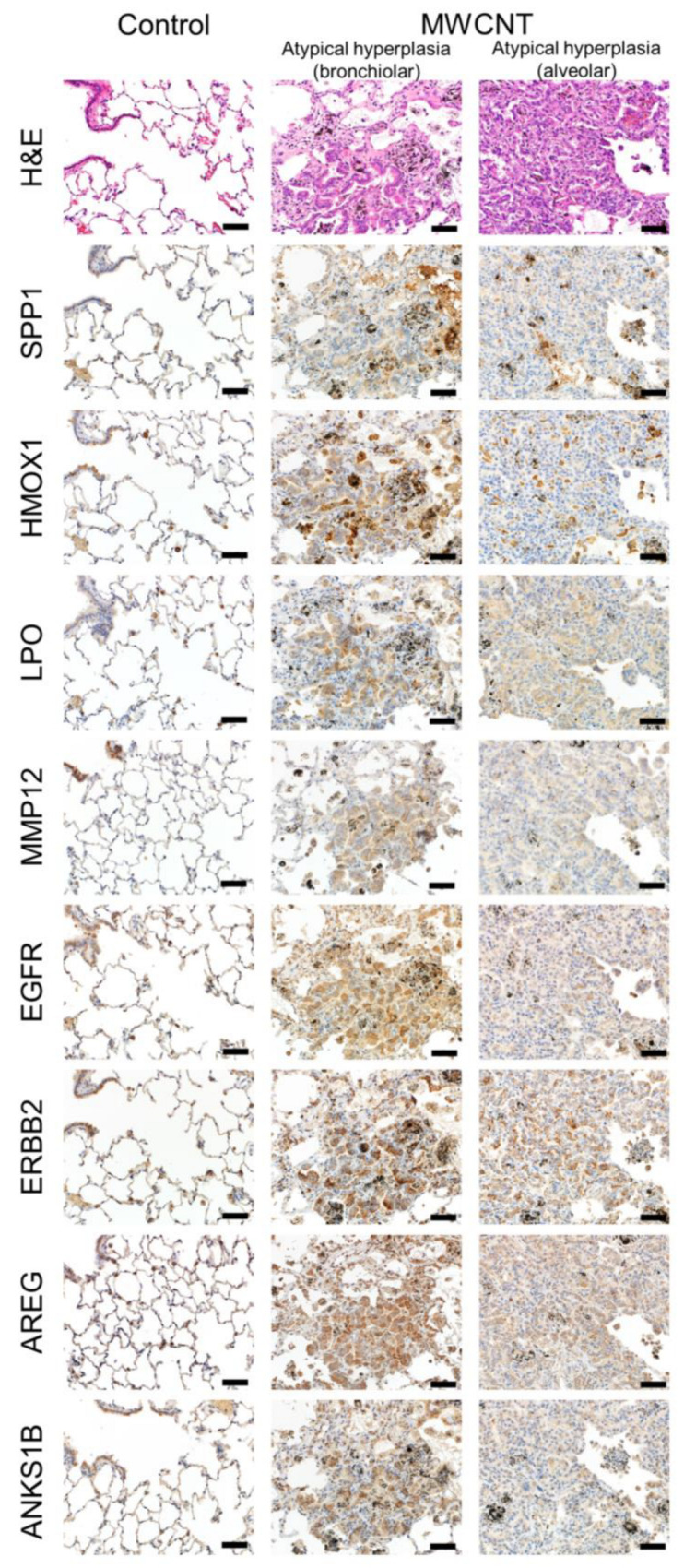
Immunohistochemistry for the selected eight proteins at Year 2. H&E (**top**) and immunohistochemical staining for the selected protein in serial sections of rat lungs of control (**left**) and MWCNT-treated groups (**center** and **right**). Protein names are shown on the left side of the photograph. The images in the center column show preneoplastic lesions consisting mainly of proliferative cells that were possibly derived from bronchiolar cells. The images in the right column show preneoplastic lesions consisting mainly of proliferative cells that were possibly derived from alveolar cells.

**Figure 10 nanomaterials-13-02105-f010:**
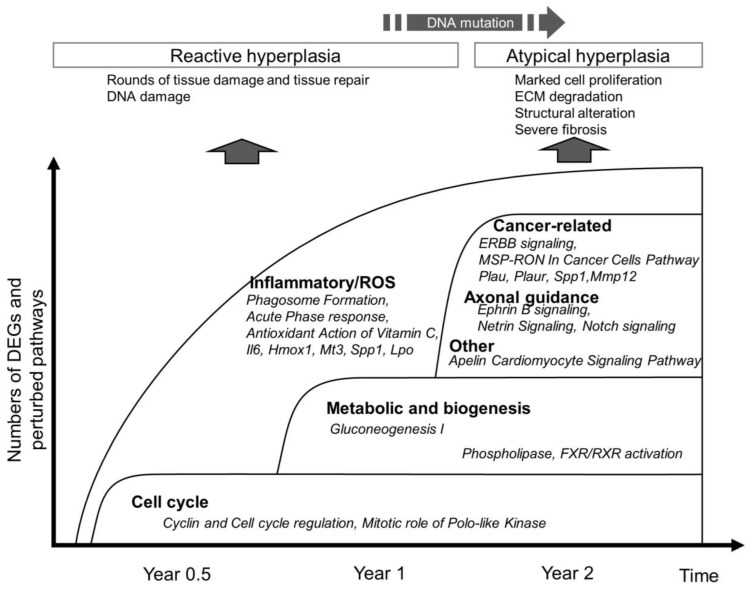
Time-course of the global gene expression pattern in lungs of rats repeatedly exposed to MWNT-7 for 2 years. A time course of the abundance of DEGs and perturbed pathways in the lungs during chronic exposure to MWNT-7 is schematically depicted, with examples of genes and pathways. Histological changes in the pulmonary epithelium and speculative biological events are also shown at the top.

**Table 1 nanomaterials-13-02105-t001:** Expression levels of the top 20 DEGs in the lung tissues of MWCNT-treated rats.

Year 0.5				Year 1				Year 2			
Upregulated		Downregulated		Upregulated		Downregulated		Upregulated		Downregulated	
Gene Symbol	Log_2_FC	Gene Symbol	Log_2_FC	Gene Symbol	Log_2_FC	Gene Symbol	Log_2_FC	Gene Symbol	Log_2_FC	Gene Symbol	Log_2_FC
*RragB*	9.32	*Faahl*	−8.32	*Mmp7*	9.57	*Rps27a_2*	−12.12	*Nlrp6_1*	9.39	*Rps27a_2*	−12.18
*Spp1*	8.98	*RT1-DMa*	−8.13	*Spp1*	9.40	*Snip1*	−8.22	*Spp1*	9.20	*Rpl39*	−11.37
*Zfp84_2*	8.65	*Snip1*	−7.69	*Mt3*	8.90	*Ndst2*	−6.19	*Ankrd34c*	8.80	*Tsen34*	−8.39
*Pla2g4e*	8.36	*Ccdc89*	−5.95	*Lpo*	7.90	*Yme1l1*	−6.18	*Mt3*	8.72	*Omd*	−7.73
*Ankrd34c*	8.06	*Sbk1*	−5.28	*Adamts18*	7.80	*Rin1*	−4.99	*Anks1b*	8.57	*Myh9*	−7.68
*Adamts18*	7.92	*Dlk1*	−5.14	*Nlrp6_1*	7.78	*Myl2*	−3.95	*Mc5r*	8.51	*Ceacam4*	−7.44
*Impad1_1*	7.79	*Myl2*	−3.98	*Defb3*	6.49	*Pnpla2*	−3.94	*Mmp7*	8.26	*Dusp11*	−7.35
*Tnf*	7.08	*Dhrs7c*	−3.08	*Retnla*	6.46	*Dlk1*	−3.69	*Lpo*	8.22	*Chid1*	−5.53
*Mmp7*	6.95	*Alpg*	−2.86	*AABR07035791.1*	6.18	*Cdh19*	−3.41	*Faim_1*	8.07	*Slc6a21*	−4.70
*Aldh18a1*	6.91	*Rbm12_1*	−2.78	*Mmp12*	5.99	*Upk1a*	−3.25	*Csap1*	8.06	*Cfc1*	−4.49
*Retnla*	6.44	*Kcna6*	−2.72	*Psca*	5.62	*Slc6a15*	−2.98	*Tpsab1*	7.86	*Acoxl*	−4.48
*Srd5a2*	6.34	*Serpinb10*	−2.66	*Sctr*	5.61	*Cct8l1_2*	−2.96	*RGD1564571*	7.47	*Dlk1*	−4.35
*Marco*	6.24	*Galnt14*	−2.62	*Slc26a4*	5.57	*Hif3a*	−2.93	*Ccl1*	7.26	*Prrt4*	−4.24
*Mt3*	6.05	*Slc6a15*	−2.55	*Reg3b*	5.51	*LOC685767*	−2.90	*Srd5a2*	7.16	*Myl2*	−4.09
*Yme1l1*	5.84	*Nhlrc4*	−2.49	*Srd5a2*	5.45	*Kcne5*	−2.89	*Mrgprx2*	6.74	*Krtap17-1*	−3.96
*Mmp12*	5.71	*Gp1bb*	−2.37	*Ocm2*	5.44	*Gpr50*	−2.88	*Sctr*	6.74	*Serpinb10*	−3.90
*Slc26a4*	5.62	*Lmod2*	−2.30	*Smtnl1*	5.44	*Clcn2*	−2.88	*Sult1c2*	6.67	*Aldh18a1*	−3.87
*Lpo*	5.47	*Sh2d6*	−2.29	*Tmem72*	5.37	*Tenm2*	−2.77	*A2m*	6.54	*Unc5d*	−3.83
*AABR07035791.1*	5.41	*Vit*	−2.23	*Ankrd34c*	5.31	*Rorb*	−2.77	*Aox4*	6.29	*Clcn2*	−3.60
*Prlhr*	5.40	*Slc6a21*	−2.22	*Orm1*	5.26	*Fbxo40*	−2.73	*Mmp12*	6.22	*Slc7a10*	−3.52

Relative expression level (Log_2_Fold Change) compared to time-matched controls.

## Data Availability

The data presented in this study are available on request from the corresponding authors.

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
