# Peer review of "Time-Course of Transcriptomic Change in the Lungs of F344 Rats Repeatedly Exposed to a Multiwalled Carbon Nanotube in a 2-Year Test"

_nanomaterials, 2023, doi:10.3390/nano13142105_

Round 1

Reviewer 1 Report

I was very pleased to read the paper submitted for review. The authors have attempted to analyse a very important aspect related to the ongoing development of nanotechnology - determining the long-term effects of exposure to one type of nanoparticle - multiwalled carbon nanotube. What is very important and worth emphasising - the experiments were planned very carefully, taking the trouble to observe the animals for a long time (2 years).
The prepared introduction comprehensively presents the current state of knowledge, including the rationale for the research undertaken.
The results are presented in a clear manner, especially the graphical presentation of the results, which makes it very easy to analyse the effects of the research work. The selection of genes analysed is not objectionable; of course, the analysis of genes directly related to protection against ROS could be extended to better characterise the adaptive changes that occur as a late effect of exposure to nanotubes. However, this is rather a suggestion for further research and not an element that detracts from the prepared article.
The research methods used are a strength of this work.
The analysis of the results obtained shows a good knowledge of the work of other teams working on similar topics, while I would like to emphasise that the authors were critical of the results obtained.
The summary presented summarises the presented results well.

The only comment - question I have for the authors is:
Was the dose of nanotubes recalculated each time there were successive repetitions of the substance administered, if the individuals showed variation in weight?
(successive administration of nanomaterials to rats).

Author Response

Thank you for your careful review.

Regarding your question, yes, we calculated the dose of the nanotube, based on individual body weight which was weighed at every administration. The average doses at each administration are summarized in our previous study (Supplementary data; Table S8).

https://static-content.springer.com/esm/art%3A10.1186%2Fs12989-022-00478-7/MediaObjects/12989_2022_478_MOESM1_ESM.pdf

Reviewer 2 Report

The article entitled “Time-course of transcriptomic change in the lungs of F344 rats repeatedly exposed to a multiwalled carbon nanotube in a 2-year test” by Hojo et al., presents a highly focused and detailed manuscript on the impact of carbon nanotubes (CNTs) on molecular mechanisms related to lung carcinogenicity over a two-year study. The authors setup a detailed rat study with appropriate numbers of animals and controls with statistical analysis. The initial animal study and pathology was previously published and focused on characterizing the multiwalled carbon nanotube and its delivery by intratracheal instillation to induce lung tumors and pleural mesotheliomas in the F344 rats. This follow-on study characterizes the transcriptional responses anchored to new histopathology using the previously published gross pathology to identify which samples for detailed molecularly pathway analysis associated with incidence lung tumors induced by low or high dose exposure to CNT’s. Other published studies have used similar protocols to look at CNT’s and their impact on model systems to include in vitro and in vivo studies.  However previous studies primarily focused on early time points, while this study collected data over a two-year period. A summary table comparing and contrasting this study to previously published work would have been helpful, as many of the same regulatory and inflammation pathways have been associated with CNT exposure. This would have also helped to further highlight the significance of the authors findings over the longer time-period studied. Overall, the article is very well written, and represents a large effort by the authors to fully characterize and statistical analyze their transcriptional findings. All the figures reflect the authors findings and support their conclusions. I believe this is an important area of research and the general readership will be interested in the manuscript and its unique findings related to CNT related carcinogenesis. I would recommend this paper for publication after addressing several specific comments to include reducing the number figures (see below).

Specific Comments:

1.     The impact of early inflammatory, ROS generation, fibrosis and DNA damage etc, could impact carcinogenesis over the lifetime of the animals. The transition between early versus later time-points is missing in the discussion and summary figure presented (Figure 11). What expression and/or specific transcripts were shared or different between this study and previously published work?  Are there tumor predictive pathways or specific of transcripts identified?

2.     The overall flow of the paper is good but could be improved by a more linear approach to presenting time with cancer progression in the result section. For instance, first discussing early time point data versus late effects in cancer progression. 

3.     Figure 4, 7 and 8 could be easily summarized in the text of the manuscript and the actual figures should be added to the supplemental figure section. 

4.     Figures 9 and 10 could be combined into a single figure with fewer example proteins to show the typical response. Additional information can be added to the supplemental section. 

5.     In Figure 10, it was not clear why MWCNT was represented twice? Is this just two different tissue sections stained? A description regarding the two columns should be included in the figure legend. 

6.     I really though Figure 11 was well thought out and should be the real focus of the discussion. This could be supported a summary table of the cancer driver genes identified within the study.

7.     HTML Links in the supplemental section need to be checked to ensure they are correct.

None

Author Response

We appreciate your helpful comments. We think we have fully addressed the concerns by adding sentences, figures and tables in the revised manuscript. All changes to our manuscript are highlighted within the document using colored text. Our answers to the comments are described point by point as follows. All referred “Page and Line” indicate them in the revised manuscript.

Reply 1. As you pointed out, we think the early events could impact carcinogenesis, thus, we state that the long-lasting inflammation may finally lead to the mutation of some genes (Page 23 Line 14 to Page 24 Line5). We have expected to find a transition (switching) of the transcriptomic signature from inflammation at early time points to other events at later time points. However, there was no clear transition, instead, the inflammation response prolonged and escalated through the experiment, and additional events occurred at later time points. We want to represent this by a stairs-like illustration. We added a sentence for an explanation of this point (Page 24 Line 5-7).

Following your comment, we added Figure S7 and Tables S8 and S9 for a comparison of data between the present study and previous studies. Because data of individual DEGs is huge, we focused on the pathways or GO terms in this comparison. These additional contents support our main texts (Page 23 Line 12-14; Page 26 Line 8-10): the inflammation related events are largely shared by previous studies, while the pathways which we are interested in (such as ERBB and guidance molecules) are specific to the present study.

Reply 2. Thank you. The way to explain results along the time course, with a comparative viewpoint early vs. late (tumorigenesis) phase, seems one of the reasonable flows. But, we should avoid to emphasize carcinogenesis with a bias, especially in Results section. In addition, it is not easy to present the results by a comparison of early and late phase as we mentioned Reply 1.

Therefore, we would like to keep the flow of the manuscript. The order of our presentation of results seems common in omics studies: i) overview of DEGs, ii) top ranked genes, iii) pathways and/or biological processes, and iv) validation experiments. In each topic (e.g., section 3.2), we did explain results according to the time axis.

Reply 3. Thank you for your comment, but, we are not attempting to remove Figures 4 and 8 from the main body. Summary of the canonical pathways (Figure 4) is one of the most important data in this study. We often see this type of figures in the main body of papers. Also, we think that the PCR validation data (Figure 8) is important and that visualization of time-course change of each gene will help readers to understand our discussion.

Regarding Figure 7, we moved it to the supplementary file following your suggestion (new Figure S6).

We will leave the decision as to whether or not to remove the Figures 4 and 8 to the editor.

Reply 4. We believe the validation data such as Figures 9 and 10 in our manuscript are so important in omics studies, and each data of immunostaining will give readers a chance to find ideas for further mechanistic studies. We are not attempting to reduce the photographs.

We will also leave the decision as to whether or not to reduce the number of photographs for immunostainings to the editor.

Reply 5. Thank you for your comment. They are not just two different tissue sections. As we described in the original figure legend, these lesions probably differ in the cell of origin. “The images in the center column show preneoplastic lesions consisting mainly of proliferative cells that were possibly derived from bronchiolar cells. The images in the right column show preneoplastic lesions consisting mainly of proliferative cells that were possibly derived from alveolar cells.”

We added an explanation on the top of the figure, which enables readers to understand the difference between them at a glans.

Reply 6. Thank you for your comment. We added a table listing lung cancer-related DEGs found in this study (Page 32 Line 16-17; new Table S10). We referred them as “lung cancer-related genes” because that includes some driver genes (oncogenic or tumor suppressor genes) and other genes thought to be possible markers for lung cancer from data of transcriptomic analyses. We think we should not restrict listed genes to “driver genes” due to lack of information of somatic mutation and epigenetic changes.

Reply 7. We checked the HTML addresses, and both link to a QIAGEN website explaining all technical terms used in IPA.

Reviewer 3 Report

Comments to the authors

The authors (Motoki et al) have developed Time-course of transcriptomic change in the lungs of F344 rats repeatedly exposed to a multiwalled carbon nanotube in a 2-year test. However, there are some points which need to be taken care of. Following are some of the comments that the authors might find useful for future submission. The manuscript should be revised before publication.

Comment: 

1.      The author should explicitly delineate the advantageous aspects of selecting the system for toxicological investigations and elucidate its potential clinical applications. Furthermore, the author's omission regarding the sterilization protocol for multi-walled carbon nanotubes (MWCNTs) is of utmost importance and warrants attention.

2.      What is the rationale behind the author's preference for carbon nanotubes (CNTs) as the subject of their long-term study?

3.      It is imperative that the author includes the appropriate abbreviations, such as single-walled carbon nanotubes (SWCNTs), upon their initial utilization within the manuscript.

4.      The author ought to incorporate a morphological analysis of CNTs, accompanied by comprehensive data regarding their size dimensions.

5.      The author should expound further on the underlying rationale for conducting a comprehensive two-year inhalation study.

6.      It is crucial for the author to include the animal approval number within the revised manuscript.

7.      Figure 7 exhibits suboptimal clarity, and the dimensions provided within the individual annotations are not discernible. Therefore, it is recommended to enhance legibility by bolding the letters or using a contrasting colour scheme.

8.      Figure 9 would benefit from the inclusion of a scale bar in each image, which would aid in accurately interpreting the depicted structures.

9.      The author should not make the fonts BOLD in between of the description.

Author Response

We appreciate your helpful comments. We think we have fully addressed the concerns by adding sentences, figures and tables in the revised manuscript. All changes to our manuscript are highlighted within the document using colored text. Our answers to the comments are described point by point as follows. All referred “Page and Line” indicate them in the revised manuscript.

Reply 1. Following your comment, we added some additional sentences in the Introduction and Discussion for explaining the advantages of our approach and the potential application of the implication of this study to humans (Page 5 Line 9-18; Page 34 Line 18 to Page 35 Line 16).

Regarding the sterilization protocol, we described it as “The MWCNT was then baked at 200 °C for 2 h in a dry heat sterilizer for the elimination of endotoxin.” in the original manuscript. Actually, we added “sterile saline solution” to the baked MWCNT (in a baked flask), and then, we sonicated the sample with an ultrasonic bath, keeping the plug of the flask closed.

Reply 2. We are interested in the chronic toxicities of nanomaterials and evaluate them using the intratracheal instillation technique. First, a shortage of animal experimental data on the lung carcinogenicity of CNT is our motivation. Until now, there is only one 2-year inhalation study by Kasai et al. (doi.org/10.1186/s12989-016-0164-2), contrasting with titanium dioxides or carbon black (IARC monograph 93). Also, potential mesothelioma inducibility is a crucial reason to prefer CNTs.

Reply 3. Thank you. We have failed to spell out some words such as SWCNT and DWCNT. We corrected them in the revision.

Reply 4. As for the characterization of the MWCNT, we explained by citing our previous paper (Hojo et al. 2022; https://doi.org/10.1186/s12989-022-00478-7) in the original manuscript. But, following your comment, we added a sentence (Page 7 Line 2-4) and a supplementary Figure showing the morphology and size distribution of MWNT-7 (new Figure S1)

Reply 5. We thought we have explained this point in the Introduction in the original manuscript. We added some additional texts to the Introduction (Page 5 Line 11-18). We also think that additional discussion (Page 34 Line18 to Page 35 Line 16), a figure (Figure S7), and tables (Tables S8 and 9) may also explain the meaning of our comprehensive analysis.

Reply 6. We added the approval numbers of our animal experiments in the Materials & Methods (Page 9 Line 7).

Reply 7. Thank you. We improved this figure by contrasting color scheme. Following the other reviewer’s comment, this figure was moved to the additional file (New Figure S6).

Reply 8. We added the scale bar in each photograph.

Reply 9. We think you mean the bold fonts for section titles such as Discussion. We simply used a template word file of Nanomaterials.

https://www.mdpi.com/journal/nanomaterials/instructions

Reviewer 4 Report

Motoki Hojo and co-authors present a quality and well-written experimental manuscript focused on time-course of transcriptomic change in the lungs of F344 rats repeatedly exposed to a multiwalled carbon nanotube in a 2-year test.

Authors aimed to obtain molecular insights into CNT-induced lung carcinogenicity.

Authors performed a transcriptomic analysis using a set of lung tissues collected from rats in a 2-year study, in which lung tumors were induced by repeated intratracheal instillations of a multiwalled carbon nanotube, MWNT-7. The RNA-seq-based transcriptome identified a large number of significantly differentially expressed genes at Year 0.5, Year 1, and Year 2. Ingenuity Pathway Analysis revealed that macrophage-elicited signaling pathways such as phagocytosis, acute phase response, and Toll-like receptor signaling were activated throughout the experimental period. At Year 2, cancer-related pathways including ERBB signaling and some axonal guidance signaling pathways such as EphB4 signaling were perturbed. qRT-PCR and immunohistochemistry indicated that several key molecules such as Osteopontin/Spp1, Hmox1, Mmp12, and ERBB2 were markedly altered and/or localized in the preneoplastic lesions, suggesting their participation in the induction of lung cancer. 

Authors found that their time-course of transcriptomic profiling of rat lungs exposed to MWNT-7 for 2 years highlighted persistent inflammation and ROS generation associated with macrophage activities. These responses to MWNT-7 exposure were evident from the earliest time point examined, 0.5 years, to the 2 years time point, and pathways associated with inflammation, ROS generation, and tissue damage demonstrated a high level of concordance with perturbed pathways identified in other omics studies with relatively short-term exposure to CNTs. Continued expose to MWNT-7 led to alterations in pathways associated with metabolic and biogenesis around 1 year after the beginning of the experiment, and caused perturbation of several cancer-related signaling and guidance molecule signaling pathways at experimental termination, suggestive of emerging of neoplastic changes in the pulmonary epithelium. 

Finally, authors conclude that their findings will increase the understanding of the molecular mechanism of CNT-induced lung carcinogenicity and can serve as a benchmark for comparison of the molecular signatures of chronic toxicity of other CNTs.

Overall, the manuscript is highly valuable for the scientific community and should be accepted for publication after the corrections are made.

==============================

Other comments:

1) Please check for typos throughout the manuscript.

2) With regards to carbon nanotubes (CNTs) – authors are kindly encouraged to cite the following article that reports the the interaction of CNTs with biomolecules, such as DNA.  This is highly relevant for transcriptomic change in various organs. DOI: 10.1016/j.jbiotec.2011.01.022

Author Response

Reply 1. Thank you so much for your careful review. We corrected all typos in the revision.

Reply 2. Thank you for your comment. The article which you suggest to cite indicates a methodology of a separation of CNTs bound to DNA. This is an interesting paper, but, we honestly don’t understand why we should cite this paper in the context of our introduction or discussion. Although we know the direct interaction of CNTs with DNA is one of the potential mechanisms of CNT-induced toxicity (Gupta et al., 2022, Nanomaterials; doi.org/10.3390/nano12101708), we discussed the carcinogenicity of MWNT-7 appeared to originate from inflammation-related, secondary toxicity (Page 24 Line 10 to Page 25 Line 1 in the revised manuscript).

Round 2

Reviewer 2 Report

I concur with the authors changes and feel the manuscript will be well received by the journals readers. I fully support publishing the article in its current form.